# FROM DATA TO REWARDS: A BILEVEL OPTIMIZATION PERSPECTIVE ON MAXIMUM LIKELIHOOD ESTIMATION

## ABSTRACT

Generative models form the backbone of modern machine learning, underpinning state-of-the-art systems in text, vision, and multimodal applications. While Maximum Likelihood Estimation has traditionally served as the dominant training paradigm, recent work have highlighted its limitations, particularly in generalization and susceptibility to catastrophic forgetting compared to Reinforcement Learning techniques, such as Policy Gradient methods. However, these approaches depend on explicit reward signals, which are often unavailable in practice, leaving open the fundamental problem of how to align generative models when only high-quality datasets are accessible. In this work, we address this challenge via a Bilevel Optimization framework, where the reward function is treated as the optimization variable of an outer-level problem, while a policy gradient objective defines the inner-level. We then conduct a theoretical analysis of this optimization problem in a tractable setting and extract insights that, as we demonstrate, generalize to applications such as tabular classification and model-based reinforcement learning.

## 1 INTRODUCTION

Generative models have become central to modern machine learning research, driving advances in text (Brown et al., 2020; DeepSeek-AI et al., 2025), image (Rombach et al., 2021; Ramesh et al., 2021), and multimodality (Zhang et al., 2024; Bai et al., 2025; Fu et al., 2025; Łajszczak et al., 2024; Yin et al., 2024) under the umbrella of "Generative AI" (*GenAI*). Their ability to synthesize realistic content has made them foundational in applications ranging from decision making (Shi et al., 2025; Kim et al., 2024; Intelligence et al., 2025) to scientific discovery (Manica et al., 2023; Lu et al., 2024).

Traditionally, such models are trained via *Maximum Likelihood Estimation* (MLE), where the parameters of the generative model are optimized to maximize the probability of observed data. This approach provides a principled framework for fitting models to large datasets and remains the backbone of many generative learning pipelines. Notably, this approach is omnipresent in today's *Large Language Models* (LLMs) through the *next token prediction* paradigm (Vaswani et al., 2023; Brown et al., 2020; DeepSeek-AI et al., 2025).

However, recent breakthroughs in LLMs research, demonstrate the limitations of MLE alone. Techniques based on Policy Gradient (PG) methods (Bellman, 1958), such as *Reinforcement Learning from Human Feedback* (Christiano et al., 2017a; Stiennon et al., 2020) and more recently *Reinforcement Learning from Verifiable Rewards* (Shao et al., 2024; DeepSeek-AI et al., 2025), have proven more effective than supervised fine-tuning at aligning models with human preferences and improving generation quality (Shenfeld et al., 2025; Lai et al., 2025; Swamy et al., 2025). These methods leverage explicit or implicit reward signals to guide training beyond likelihood objectives.

Yet, in many practical scenarios, explicit reward functions are unavailable. Instead, we often possess high-quality datasets on which we would like to align our models. This raises a key question:

*Can we learn an **implicit reward function** from **unlabeled data**, and exploit the well-developed
**policy optimization** literature to train models **more effectively** than with MLE?*

In this paper, we propose the following contributions toward addressing this question:

- **Bilevel optimization perspective on MLE:** We reinterpret the MLE training objective as a *Bilevel Optimization* (Bi-O) problem, where the outer-level problem optimizes over the reward function, while the inner-level problem is defined by a PG objective with respect to the model parameters.
- **Theoretical analysis:** We study this formulation under a Gaussian data distribution with the reward given by a negatively scaled distance in the output space, deriving insights into the theoretically optimal parameters of the reward function.
- **Practical algorithms:** Guided by the theoretical analysis and leveraging implicit differentiation solvers, we propose two practical algorithms for addressing the bilevel optimization problem. We evaluate these algorithms on two MLE applications: tabular classification and model-based reinforcement learning.

The remainder of the paper is organized as follows. Section 2 situates our work within the relevant literature, and Section 3 introduces the problem setup and motivates our approach. In Section 4, we address the bilevel optimization problem in the Gaussian case, while Section 5 considers the general setting using implicit differentiation. We then present experimental results in Section 6 and conclude with a discussion in Section 7.

## 2 RELATED WORK

**PG vs MLE for Generative models.** Generative models aim to capture the underlying distribution of observed data, with the goal of synthesizing realistic samples afterwards, e.g. text generation (Brown et al., 2020) and image generation (Rombach et al., 2021; Ramesh et al., 2021). Many of the existing generative modeling approaches as Autoregressive models (Radford & Narasimhan, 2018; Vaswani et al., 2023; Radford et al., 2019), Variational AutoEncoders (Kingma & Welling, 2013; Higgins et al., 2017), Generative Adversarial Networks (Goodfellow et al., 2014; Arjovsky et al., 2017), Diffusion Models (Sohl-Dickstein et al., 2015; Rombach et al., 2021), can be framed through the lens of MLE or its approximations. However, and especially in the context of sequence generation, MLE in autoregressive models has been proven to suffer from compounding errors and exposure bias, among other problems (Tan et al., 2019; Bahdanau et al., 2017; Ranzato et al., 2016; Bengio et al., 2015; Venkatraman et al., 2015; Benechehab et al., 2024). As an alternative approach, PG methods have emerged as a more effective way to sample the output space when a reward function is available (Bahdanau et al., 2017). Beyond vanilla PG, more sophisticated methods have been developed, such as Reward-Augmented Maximum Likelihood (Norouzi et al., 2016; Volkovs et al., 2011), where a reward-based stationary sampling distribution is defined, Softmax Policy Gradient (Ding & Soricut, 2017), an intermediate approach between sampling the model and sampling a reward-based distribution, and MIXER (Ranzato et al., 2016), a scheduling approach that gradually transitions from MLE to PG using the REINFORCE algorithm (Williams, 1992). Besides autoregressive models, policy gradient methods have also been used to train (or finetune) Diffusion models (Black et al., 2024; Uehara et al., 2024; Zekri & Boullé, 2025), and GANs (Paria et al., 2017; Yu et al., 2017).

**Reward models.** Policy Gradient methods constitute one class of algorithms for solving *Markov Decision Processes* (MDP) (Bellman, 1958), the central formalism underpinning the RL field. Training generative models with PG methods builds on the formulation of the task as an MDP. In this setting, the reward function plays a pivotal role. The most direct way of learning a reward model is via supervised learning from past interactions, as done in *Model-based Reinforcement Learning* (Chua et al., 2018; Janner et al., 2019; Yu et al., 2020; Hafner et al., 2021; Kégl et al., 2021; Benechehab et al., 2025). Beyond the supervised approach, several other paradigms for reward learning have been developed. *Learning from Demonstrations* includes Inverse RL methods (Abbeel & Ng, 2004; Ziebart et al., 2008; Finn et al., 2016a;b) that learn a reward model $R_\theta$ under which demonstrations of the form $(s, a, s_{\text{next}})$ are optimal. Another paradigm, *Learning from Goals*, defines the reward function with respect to reaching a goal $g$ in the state space $\mathcal{S}$ (Liu et al., 2022). In this setting, goal attainment has been modeled in terms of spatial distances (Nachum et al., 2018; Mazzaglia et al., 2024), temporal distances (Hartikainen et al., 2020; Wang et al., 2025), and semantic similarity (Sontakke et al., 2023; Fan et al., 2022). The *Learning from Preferences* approach relies on transforming preference data of the form $(\tau_0 \succ \tau_1)$, where $\tau_i$ is a trajectory $(s_1, a_1, \ldots, s_{|\tau_i|}, a_{|\tau_i|})$ and $\succ$ is a preference

relationship, into a reward model using the *Bradley-Terry* model (Bradley & Terry, 1952). Reward models learned from preference data have enabled significant progress in *post-training* generative models (Kim et al., 2023; Touvron et al., 2023; Rafailov et al., 2023; Song et al., 2024). Starting with *InstructGPT* (Ouyang et al., 2022), this approach has become a standard for improving targeted aspects of LLMs, e.g. safety (Dai et al., 2024), as well as for applications such as mathematical reasoning (Xin et al., 2025; Shao et al., 2024; Luong et al., 2024) and code generation (DeepSeek-AI et al., 2025).

**Bilevel optimization.** Bilevel Optimization (Bi-O) was originally introduced in economics and game theory by von Stackelberg (1934) to model hierarchical decision-making problems between a leader and a follower. More broadly, Bi-O offers a framework for addressing problems with hierarchical structures, where the task is to optimize two interdependent objective functions: an *inner-level* objective and an *outer-level* objective. In machine learning, Bi-O was first applied to feature selection (Bennett et al., 2006) and was later extended to a wide spectrum of applications, including hyperparameter optimization (Mackay et al., 2019; Franceschi et al., 2017; Pedregosa, 2016), reinforcement learning (Hong et al., 2022; Nikishin et al., 2021), and meta-learning (Franceschi et al., 2018). Various Bi-O solvers have been proposed to address different regularity conditions on the inner- and outer-level objectives. Among these, *automatic differentiation*-based approaches compute gradients of the outer-level objective by differentiating through the iterative steps of the inner-level optimization algorithm (Wengert, 1964; Linnainmaa, 1976; Domke, 2012; Franceschi et al., 2017). In parallel, *implicit differentiation* methods (Bengio, 2000) leverage the implicit differentiation theorem to approximately estimate the gradient of the outer loss by solving a linear system (Pedregosa, 2016; Chen et al., 2021; Ji et al., 2021; Arbel & Mairal, 2022). Beyond alternating methods, Dagréou et al. (2024) introduce a framework where inner- and outer-level variables evolve jointly within a single training loop. Bi-O has also been generalized to functional settings (Petrulionyte et al., 2024), where the inner-level optimization is carried out over functions in infinite-dimensional spaces. In the context of generative models, some approaches enhance the training efficiency of energy-based latent variable models through bilevel formulations (Bao et al., 2020; Kan et al., 2022), while Xiao et al. (2025) propose a bilevel framework for tuning hyperparameters and noise schedules in diffusion models.

**Bilevel Reinforcement Learning.** Bilevel RL optimizes an outer-level objective, often a reward function or alignment signal, while an inner loop learns a policy under that objective. This framework has been applied in areas such as reward shaping (Zou et al., 2019) or RLHF (Christiano et al., 2017b; Xu et al., 2020). The closest work to ours is (Zeng et al., 2022), which combines MLE with inverse RL methods, however they focus on control tasks while we aim at providing a general framework for any data modality.

## 3 PRELIMINARIES

In Section 3.1, we motivate learning reward functions from data and outline when PG methods may outperform MLE. We then formally define the problem setup in Section 3.2.

### 3.1 MOTIVATION

In Reinforcement Learning, PG methods are traditionally viewed as producing unbiased yet high-variance gradient estimates, especially in long-horizon or high-dimensional tasks (Greensmith et al., 2001). In contrast, MLE has historically served as the dominant paradigm in supervised learning and probabilistic modeling (Akaike, 1998). However, in the current era of large pretrained models and advanced RL algorithms, these limitations have become less restrictive, giving rise to many cases where PG methods are more advantageous than MLE.

A first phenomenon is the *mismatch between training objectives and evaluation metrics*. In sequence prediction, for instance, evaluation scores such as BLEU or ROUGE do not decompose into token-level likelihoods. While for the widely used autoregressive models MLE is restricted to maximizing token-level likelihoods, PG methods directly optimize sequence-level rewards and naturally account for this discrepancy (Norouzi et al., 2016; Ding & Soricut, 2017; Ranzato et al., 2016).

Another key phenomenon is *catastrophic forgetting*. When adapting large language models to downstream tasks through post-training, it is often desirable to preserve prior knowledge while

specializing to new distributions. Recent studies (Shenfeld et al., 2025; Lai et al., 2025; Swamy et al., 2025) suggest that on-policy RL fine-tuning achieves this balance more effectively than supervised fine-tuning, since its updates converge to solutions closest in KL divergence to the original policy.

Taken together, these observations motivate our approach: rather than maximizing the likelihood directly, we propose a general framework that interprets data signals as reward functions, thereby also enabling PG optimization.

## 3.2 Problem setup

Let $(\Omega, \mathcal{F}, \mathbb{P})$ be a probability space, and let $X : \Omega \to \mathcal{X}$ and $Y : \Omega \to \mathcal{Y}$ be two random variables, with $\mathcal{X} \subseteq \mathbb{R}^m$ and $\mathcal{Y} \subseteq \mathbb{R}^n$, where $(n, m) \in \mathbb{N}_\star^2$. Consider a maximum likelihood estimation problem where we observe $N$ i.i.d realizations $\mathcal{D} = \{(\mathbf{x}_i, \mathbf{y}_i)\}_{i=0}^N$ from a fixed unknown distribution over $\mathcal{X} \times \mathcal{Y}$. The goal is to model the conditional distribution $Y|X \sim q$ using a parametric model $\widehat{Y}|X \sim \hat{p}_\theta$ where $\theta \in \Theta := \mathbb{R}^{d_\theta}$ are parameters spanning a finite dimensional space with dimension $d_\theta$. In the MLE formalism, we optimize the parameters $\theta$ by maximizing the log-likelihood, equivalently seen as a Kullback-Leibler divergence minimization (Akaike, 1998) (denoted as $d_{\mathrm{KL}}$):

$$\theta^\star = \arg\min_{\theta \in \Theta} \mathbb{E}_X[d_{\mathrm{KL}}(q(\cdot|X)||\hat{p}_\theta(\cdot|X))] = \arg\max_{\theta \in \Theta} \mathbb{E}_X \mathbb{E}_{Y|X \sim q}[\log \hat{p}_\theta(Y|X)] \qquad \text{(MLE)}$$

A parallel approach, based on reinforcement learning, consists in maximizing a reward function $r : \mathcal{Y} \times \mathcal{Y} \to \mathbb{R}$ that evaluates the quality of generated $\hat{\mathbf{y}}$ against the true observations $\mathbf{y}$, resulting in the Policy Gradient (PG) objective. Here we state the entropy-regularized PG objective, a variant that is commonly considered in RL algorithms (Haarnoja et al., 2017; 2018; Wen et al., 2024):

$$\theta^\star = \arg\max_{\theta \in \Theta} \mathbb{E}_X \mathbb{E}_{Y|X \sim q}\Big[\mathbb{E}_{\widehat{Y}|X \sim \hat{p}_\theta}\Big[r(\widehat{Y}, Y)\Big] + \lambda \mathrm{H}(\hat{p}_\theta)\Big], \qquad \text{(PG)}$$

where $\lambda > 0$ is a parameter controlling the strength of the regularization, and H denotes the entropy.

In this work, we ask whether the reward function itself can be seen as an optimization variable $r$ over a Hilbert space $\mathcal{H}$. The *optimal reward function* is then determined based on the MLE objective, which now represents the outer-level of the following bilevel optimization problem:

$$\max_{r \in \mathcal{H}} \mathbb{E}_X \mathbb{E}_{Y|X \sim q}\Big[\log \hat{p}_{\theta_r^\star}(Y|X)\Big] \quad \text{s.t.} \quad \theta_r^\star = \arg\max_{\theta \in \Theta} \mathbb{E}_X \mathbb{E}_{Y|X \sim q}\Big[\mathbb{E}_{\widehat{Y}|X \sim \hat{p}_\theta}[r(Y', Y)] + \lambda \mathrm{H}(\hat{p}_\theta)\Big]$$
$$\text{(Bi-O)}$$

## 4 Solving Bi-O in a tractable case

The objective of this section is to analyze the bilevel optimization problem Bi-O under specific assumptions on the data-generating distribution and the reward parametrization, in which both the inner- and outer-level problems admit closed-form solutions.

## 4.1 Theoretical analysis

We start our analysis by stating the following assumptions, which will prove useful in the establishment of our main results:

**Assumption 4.1** (Gaussian density model). We assume that both the true conditional density $q$ and the model density $\hat{p}_\theta$ are Gaussian distributions with linear mean functions and fixed covariance matrices:
$$Y \mid X \sim q := \mathcal{N}(\Lambda X, \Sigma), \quad \widehat{Y} \mid X \sim \hat{p}_\theta := \mathcal{N}(AX, B),$$
where $\Lambda \in \mathbb{R}^{n \times n}$, $\Sigma \in \mathrm{S}_n^{++}(\mathbb{R})$[1], and $\theta := (A, B) \in \Theta := \mathbb{R}^{n \times n} \times \mathrm{S}_n^{++}(\mathbb{R})$.

---

[1]We denote by $\mathrm{S}_n^{++}(\mathbb{R})$ and $\mathrm{S}_n^+(\mathbb{R})$ the sets of real symmetric positive definite and positive semidefinite $n \times n$ matrices, respectively.

**Assumption 4.2** (Reward model). Let $U \in S_n^{++}(\mathbb{R})$, we define the reward model as the following quadratic form: $\forall (\widehat{Y}, Y) \in \mathbb{R}^n \times \mathbb{R}^n, \quad r_U(\widehat{Y}, Y) = -(\widehat{Y} - Y)^T U(\widehat{Y} - Y)$.

We first notice that this choice of parametrization is valid as the resulting reward function is maximized in $Y$. Furthermore, this parametrization enables that for any $U \in \mathbb{R}^{n \times n}$, $r_U$ is an element of the Hilbert space $\mathcal{H}$ of square-integrable real-valued functions with a weighted measure. We refer the interested reader to Appendix A.2 for a technical definition of $\mathcal{H}$ and a proof of this statement. We now state the main results, showcasing closed-form solutions of the Bi-O problem under the previous assumptions.

**Proposition 4.3.** *Under assumptions 4.1 and 4.2, the Bi-O problem has exactly one solution that writes:*

$$U^\star = \tfrac{\lambda}{2} \Sigma^{-1}.$$

The proof of proposition 4.3 is deferred to Appendix A.1.

**Corollary 4.4.** *If we assume that, $B = \sigma^2 I_n$, the set of solutions to the Bi-O problem is characterized by:*

$$U^\star \in F_{\lambda, \Sigma} := \left\{ U \in S_n^{++}(\mathbb{R}) \,\middle|\, \operatorname{Tr}(U) = \frac{\lambda n^2}{2 \operatorname{Tr}(\Sigma)} \right\}.$$

Note that, for any given $\lambda > 0, \Sigma \in S_n^{++}(\mathbb{R})$, $F_{\lambda, \Sigma} \neq \varnothing$ since $\frac{\lambda n}{2 \operatorname{Tr}(\Sigma)} I_n \in F_{\lambda, \Sigma}$ which corresponds to reward functions we consider for the empirical experiments in Section 4.2.

**Interpretation as Mahalanobis distance.** The optimal reward function obtained by the Proposition 4.3 is linked to the inverse covariance matrix, leading to an interpretation as the negative squared Mahalanobis distance (Mahalanobis, 1936) between $\widehat{Y}$ and a Gaussian centered at $Y$. Since $Y$ is an unbiased estimator of $\Lambda X$, this reflects distance to the underlying distribution. Thus, noisier data reduce penalization for deviations, while the scaling factor $\lambda$ balances reward maximization and entropy regularization.

**Interpretation as reverse KL minimization.** An interesting observation arises when substituting the optimal reward parametrization $U^\star$ into the inner-level objective: the PG formulation becomes equivalent to minimizing the reverse KL divergence between the model distribution $\hat{p}_\theta$ and the data-generating distribution $q$. This connection provides an explanation for the empirical results presented in the next section. We therefore state it formally as a corollary, with the proof deferred to Appendix A.1.5:

**Corollary 4.5.** *Under the assumptions of Proposition 4.3, the optimal parameters $\theta_{U^\star}^\star$ obtained from the lower-level problem with $U^\star = \tfrac{\lambda}{2} \Sigma^{-1}$ minimize the reverse KL divergence between $\hat{p}_\theta$ and $q$, (i.e)* $\theta_{U^\star}^\star = \arg\min_{\theta \in \Theta} \mathbb{E}_X \left[ d_{KL} \left( \hat{p}_\theta(\cdot|X) \,\|\, q(\cdot|X) \right) \right].$

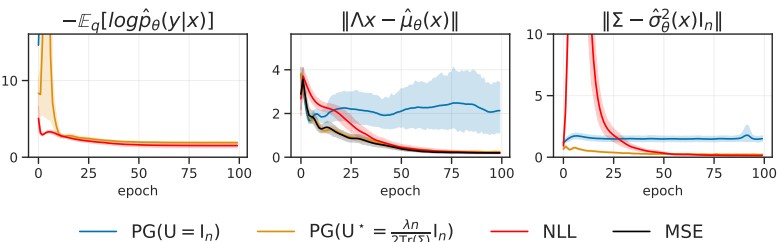

Figure 1: **Synthetic data experiment.** The PG loss when paired with the optimal reward function matches the NLL-trained baseline in terms of NLL (left panel), all while having faster convergence in terms of moment matching (center and right panels for the mean and variance, respectively).

## 4.2 EMPIRICAL VALIDATION

In this section, we evaluate the theoretical results from Section 4.1. To this end, we generate synthetic observed data that satisfy Assumption 4.1: $\mathcal{D} = \{(\mathbf{x}_i, \mathbf{y}_i)\}_{i=0}^N$, where $\mathbf{x}_i \sim \mathcal{U}([-5,5]^n)$ ($\mathcal{U}$ denotes the uniform distribution), and $\mathbf{y}_i \sim q(.|\mathbf{x}_i) := \mathcal{N}(\Lambda \mathbf{x}_i, \Sigma)$ with diagonal covariance matrix $\Sigma = \beta^2 \mathrm{I}_n$ and $\beta > 0$. For the model $\hat{p}_\theta$, we relax the linearity and homoscedasticity assumptions by considering a neural network that parametrizes a Gaussian distribution, in which both the mean function and the diagonal covariance matrix depend on the input: $\hat{p}_\theta(.|\mathbf{x}_i) := \mathcal{N}(\mu_\theta(\mathbf{x}_i), \sigma_\theta^2(\mathbf{x}_i) \mathrm{I}_n)$. We compare baselines trained with *negative log-likelihood* (NLL) and *mean squared error* (MSE) losses against PG variants, using either a negative squared distance reward $\mathrm{U} = \mathrm{I}_n$ or the optimal reward function derived in Corollary 4.4 with $\mathrm{U}^\star = \frac{\lambda n}{2 \operatorname{Tr}(\Sigma)} \mathrm{I}_n$.

Fig. 1 shows validation NLL and moment-matching errors (mean and covariance) over training. Consistent with theory, we observe that adjusting the reward function with the optimal matrix $\mathrm{U}^\star$ yields a learning curve nearly identical to the NLL baseline (yellow and red curves in the left panel of Fig. 1). Moreover, the PG variant with the optimal matrix converges faster than the NLL baseline in matching the moments of the data-generating distribution (center and right panel). Finally, we note that the vanilla PG method (with $\mathrm{U} = \mathrm{I}_n$) suffers from a diverging NLL due to the variance shrinking to zero for some values of $\lambda$ which leads to numerical instabilities.

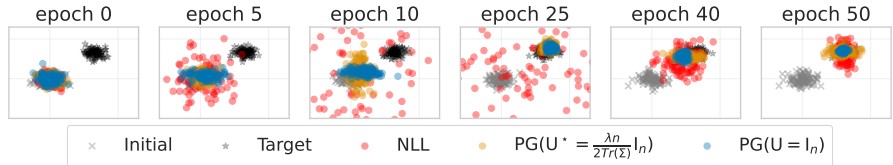

Figure 2: **Learned distributions comparison on a single data point.** The PG loss paired with the optimal reward function in Corollary 4.4 shows optimal convergence, even when compared with the baseline directly optimizing the NLL.

To gain further insight, Fig. 2 shows the evolution of the learned distributions for a single training data point. In this illustrative example, the PG variant with the optimal reward displays the most natural behavior in fitting the target distribution, unlike the NLL baseline, which initially causes the variance to increase sharply before reducing it to match the target variance. This behavior can be explained by Corollary 4.5 since minimizing $d_{KL}\left(\hat{p}_\theta(\cdot|X) \,\|\, q(\cdot|X)\right)$ is known to induce mode seeking behavior.

## 5 SOLVING BI-O IN GENERAL

In contrast to the previous section, where we assumed access to the data-generating distribution and provided a closed-form solution to problem Bi-O, real-world applications typically do not satisfy such assumptions. Consequently, solving the bilevel optimization problem Bi-O by directly optimizing the outer objective offers a more general approach applicable to a broader class of problems.

Bilevel optimization solvers can generally be divided into three categories. Explicit gradient methods treat the gradient update as a differentiable mapping and backpropagate through the unrolled optimization path of the inner-level problem (Franceschi et al., 2017). Gradient-free methods instead rely on evolutionary strategies, optimizing the outer objective while considering the inner problem as a black-box function (Song et al., 2020; Feng et al., 2021). Finally, implicit differentiation methods leverage the implicit function theorem to reformulate gradient estimation as the solution to a linear system (Dagréou et al., 2024; Petrulionyte et al., 2024).

In this work, we focus on implicit differentiation, as explicit gradient methods often encounter memory issues from storing long computational graphs, while gradient-free approaches are generally limited by the curse of dimensionality.

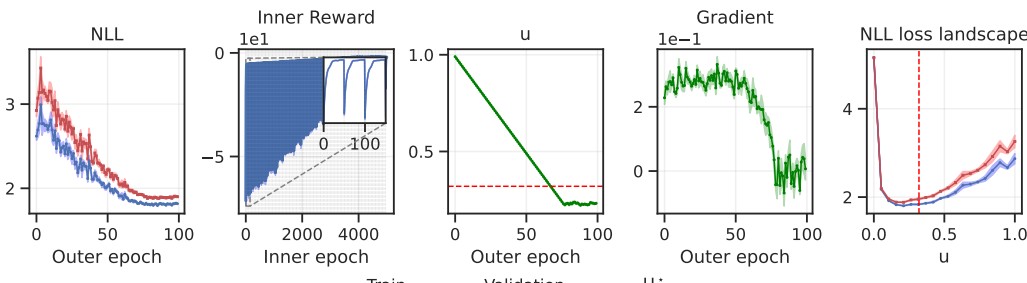

Figure 3: **Implicit differentiation solver on synthetic data experiment.** From left to right: outer loss (NLL), inner reward optimization loop, trajectory of the reward parameter $\mathrm{u}$, gradient of the outer loss with respect to $\mathrm{u}$, outer loss landscape.

### 5.1 IMPLICIT DIFFERENTIATION

Consider a reward parametrization $r_\phi$ with $\phi \in \Phi := \mathbb{R}^{d_\phi}$, where $d_\phi$ denotes the dimension of the reward parameter space. The optimization of the outer-level problem can thus be restricted to the Hilbert space of reward functions spanned by parameters $\phi \in \Phi$ (see the appendix for an explicit construction in the case of the Mahalanobis parametrization). Within this setup, implicit differentiation treats the solution of the inner problem, $\theta^\star$, as an implicit function of $\phi$ and allows one to compute the best-response derivatives $\nabla_\phi \theta^\star(\phi)$ analytically via the implicit function theorem.

To proceed, we define an operator $\mathrm{f} : \Phi \times \Theta \to \Theta := \mathbb{R}^{d_\theta}$ whose roots characterize the inner-level optimal solution $\theta^\star(\phi)$. That is, for all $\phi \in \Phi$, we have $\mathrm{f}(\phi, \theta^\star(\phi)) = 0$. Leveraging this property, the derivative of interest $\nabla_\phi \theta^\star(\phi)$ can be determined by solving for $\nabla_\phi \mathrm{f}(\phi, \theta^\star(\phi)) = 0$:

$$\forall \phi \in \Phi, \quad \nabla_\theta f(\phi, \theta^\star(\phi)) \nabla_\phi \theta^\star(\phi) + \nabla_\phi f(\phi, \theta^\star(\phi)) = 0, \tag{1}$$

where $\nabla_\phi \theta^\star(\phi)$ is obtained by solving the linear system in Eq. (1), enabling gradient descent on the outer problem via the chain rule.

In our bilevel optimization formulation, the operator $\mathrm{f}$ arises naturally from the fixed-point characterization of the gradient update: $\mathrm{f}(\phi, \theta) = \theta + \alpha \nabla_\theta \mathcal{L}_{\text{in}}(\phi, \theta) - \theta = \alpha \nabla_\theta \mathcal{L}_{\text{in}}(\phi, \theta)$ where $\mathcal{L}_{\text{in}}$ is the inner-level objective and $\alpha$ is a learning rate. Under this definition, the first-order optimality condition holds whenever the inner-level optimization converges to a local minimum $\theta^\star(\phi)$, where the gradient vanishes, which we assume is a plausible hypothesis given any modern stochastic optimizer (e.g. Adam (Kingma & Ba, 2017)).

### 5.2 EMPIRICAL VALIDATION

In practice, we use TorchOpt (Ren et al., 2023), a python package that enables differentiable optimization solvers that can be integrated with pytorch based neural network implementations. Precisely, we run the implicit differentiation-based solvers using the Conjugate Gradient algorithm for the linear system resolution, as in (Rajeswaran et al., 2019). We now compare the obtained results, in the same setup as Section 4.2, to get insights into the effectiveness of this kind of bilevel optimization solvers against MLP-based policies.

Fig. 3 presents the results of running an implicit differentiation solver for 100 outer iterations, each with 50 inner iterations, and a learning rate of $10^{-2}$ for both optimization loops. The outer optimization variable is a single parameter (central panel) initialized at 1, which defines the diagonal Mahalanobis matrix for the reward: $\mathrm{u} > 0$ s.t. $\mathrm{U} = \mathrm{u} \cdot \mathrm{I}_n$. The leftmost panel illustrates the evolution of the outer loss (NLL evaluated on the optimal policy from the inner PG loop), showing clear improvement relative to the initialization at 1 (which corresponds to the Euclidean distance). Additionally, the optimization parameter $\mathrm{u}$ converges to a value close to the theoretical optimum $\mathrm{u}^\star = \frac{\lambda n}{2 \operatorname{Tr}(\Sigma)}$, as derived in Corollary 4.4. This convergence is further supported by the far-right panel, which plots the outer loss landscape as a function of the reward parameter $\mathrm{u}$, revealing a roughly convex landscape with a global minimum near the theoretical optimum. These results validate our

intuition from the tractable case discussed in Section 4, even in the more general setting of MLP-based policies and stochastic optimization solvers within the implicit differentiation framework.

# 6 APPLICATIONS

The goal of this section is to use intuition gained from the previous analysis to derive practical algorithms that we can validate on common NLL tasks from the literature.

---

**Algorithm 1** PG($U_{he}^\star$) - heuristic

**Input:** Data $\mathcal{D} = \{(\mathbf{x}_i, \mathbf{y}_i)\}_{i=0}^N$, model $\hat{p}_\theta$, $\lambda$
**1.** Estimate cov matrix $\hat{\Sigma} = \text{cov}(\{(\mathbf{y}_i)\}_{i=0}^N)$
**2.** loss $\leftarrow$ PG($U_{he}^\star(\lambda, \hat{\Sigma})$)
**3.** train_policy($\hat{p}_\theta, \mathcal{D}, \text{loss}$)
**Return:** learned model $\hat{p}_{\theta^\star}$

---

**Algorithm 2** PG($U_{im}^\star$) - implicit differentiation

**Input:** Data $\mathcal{D} = \{(\mathbf{x}_i, \mathbf{y}_i)\}_{i=0}^N$, model $\hat{p}_\theta$, $\lambda$
**1.** $U_{im}^\star \leftarrow$ imp_diff_solver($\mathcal{D}, \lambda$)
**2.** loss $\leftarrow$ PG($U_{im}^\star$)
**3.** train_policy($\hat{p}_\theta, \mathcal{D}, \text{loss}$)
**Return:** learned model $\hat{p}_{\theta^\star}$

---

First, we build on the theoretical analysis in Section 4.1 to suggest a realistic way to estimate the optimal reward parametrization $U^\star$ derived in. The main challenge with this approach lies in estimating the covariance matrix-dependent term. As stated in Algorithm 1, we propose an approximate approach that estimates an empirical covariance matrix $\hat{\Sigma}$ from the training data. Secondly, we use the implicit differentiation-based bilevel solver to provide a gradient-based approach (Algorithm 2). Such an approach, is more general as it's not sensitive to the estimation error on the covariance matrix, nor requires the validity of the assumptions under which we derive our theoretical results.

In the following, we use both Algorithms 1 and 2 to benchmark our method against vanilla PG and NLL losses in two real-world applications: tabular classification, and model-based reinforcement learning. Note that, in the experiments, we are effectively solving the inner-level problem of the Bi-O formulation, while substituting the reward function either with the optimal matrices $U_{he}^\star$ and $U_{im}^\star$, or with the identity $I_n$ for the squared-distance baseline.

## 6.1 TABULAR CLASSIFICATION

We evaluate our framework on several tabular classification datasets from the UCI repository (Wang, 2023). Specifically, we train a multiclass logistic regression model with the PG loss, where the reward is defined as in Theorem 4.2. We consider both multiclass (Poker) and imbalanced binary classification (Credit default). In the case of unbalanced datasets, accuracy alone can be misleading, in which case we additionally report the Area Under the Curve (AUC).

Table 1: credit_default: mean $\pm$ *variance* of AUC over 3 runs.

| Method | $\mathbf{AUC}_{/10^{-2}} \uparrow$ |
|---|---|
| **NLL** | $70.5 \pm 5.04 \times 10^{-5}$ |
| $\mathbf{PG}(I_n)$ | $57.7 \pm 6.32 \times 10^{-3}$ |
| $\mathbf{PG}(U_{he}^\star)$ | $\mathbf{71.3} \pm 1.00 \times 10^{-8}$ |

Table 2: Accuracy (mean $\pm$ *variance*)

| Dataset | Method | $\mathbf{Accuracy}_{/10^{-2}} \uparrow$ |
|---|---|---|
| *Credit default* | **NLL** | $79.8 \pm 1.21 \times 10^{-4}$ |
| | $\mathbf{PG}(I_n)$ | $75.5 \pm 2.34 \times 10^{-4}$ |
| | $\mathbf{PG}(U_{he}^\star)$ | $\mathbf{82.0} \pm 2.50 \times 10^{-7}$ |
| *Poker* | **NLL** | $48.6 \pm 1.23 \times 10^{-5}$ |
| | $\mathbf{PG}(I_n)$ | $38.2 \pm 3.92 \times 10^{-4}$ |
| | $\mathbf{PG}(U_{he}^\star)$ | $\mathbf{52.4} \pm 1.00 \times 10^{-6}$ |

Table 2 shows the accuracy results across 3 datasets, while Table 1 extends this to AUC for the binary classification task. On the imbalanced Credit default dataset, accuracy is high across methods but AUC reveals that $\mathbf{PG}(U_{im}^\star)$ better separates classes. The Poker dataset remains challenging for all methods, yet $\mathbf{PG}(U_{im}^\star)$ still provides the best performance.

## 6.2 MODEL-BASED REINFORCEMENT LEARNING

Model-Based Reinforcement Learning (MBRL) addresses the supervised learning problem of estimating the (possibly stochastic) transition function of a MDP. Typically, we assume access to data of the form $\mathcal{D} = \{(s_t^i, a_t^i, s_{t+1}^i)\}_{i=0}^N$, consisting of trajectories of states $s$ and actions $a$ collected by an unknown policy. The goal is to approximate the next-state distribution $S_{t+1} \mid S_t, A_t \sim q$. In practice, the dynamics model is often a Gaussian probabilistic model trained via log-likelihood (Chua et al., 2018; Janner et al., 2019), which makes it directly applicable to our experimental setup. We consider three D4RL (Fu et al., 2021) *HalfCheetah* tasks, each from a different data-collecting policy: *simple*, *medium*, and *expert*, accessible through the *Minari* project (Younis et al., 2024). All models train for 400 epochs with Adam optimizer (learning rate = $10^{-3}$) and $\lambda = 1$.

Table 3 presents MSE and NLL results across the different losses under evaluation. As expected, NLL-optimized models achieve the strongest performance on NLL. However, consistent with the synthetic data experiments in Section 4.2, we observe PG with the optimal reward heuristic $\mathbf{PG}(\mathrm{U}_{\mathrm{he}}^\star)$ delivers significant NLL improvements compared to PG with the negative squared-distance reward $\mathbf{PG}(\mathrm{I}_n)$. Moreover, the optimal reward obtained from the implicit differentiation solver $\mathbf{PG}(\mathrm{U}_{\mathrm{im}}^\star)$ achieves the second-best NLL performance while achieving best MSE, an important property in the context of MBRL, particularly when using deterministic planners.

These findings support our intuition that PG methods with an optimal reward can enhance NLL (and also MSE), as guaranteed by our bilevel optimization framework. It is worth emphasizing that, in the context of MBRL, the metric of ultimate interest is the policy performance derived from these models, typically quantified by the return (i.e., the discounted cumulative reward up to the task horizon). We defer exploration of this direction to future work, as the present paper concentrates on the MLE task.

| Task | Metrics | |
| --- | --- | --- |
| | $\mathbf{MSE}_{/10^{-2}} \downarrow$ | $\mathbf{NLL}_{/10^{-2}} \downarrow$ |
| **NLL** | | |
| simple | $425 \pm 3$ | $\mathbf{47 \pm 1}$ |
| medium | $459 \pm 3$ | $\mathbf{73 \pm 1}$ |
| expert | $539 \pm 3$ | $\mathbf{49 \pm 1}$ |
| $\mathbf{PG}(\mathrm{I}_n)$ | | |
| simple | $199 \pm 1$ | $528 \pm 18$ |
| medium | $241 \pm 4$ | $796 \pm 70$ |
| expert | $\mathbf{174 \pm 3}$ | $420 \pm 24$ |
| $\mathbf{PG}(\mathrm{U}_{\mathrm{he}}^\star)$ | | |
| simple | $230 \pm 1$ | $267 \pm 1$ |
| medium | $274 \pm 1$ | $286 \pm 1$ |
| expert | $\underline{198 \pm 1}$ | $290 \pm 1$ |
| $\mathbf{PG}(\mathrm{U}_{\mathrm{im}}^\star)$ | | |
| simple | $\mathbf{190 \pm 1}$ | $\underline{208 \pm 1}$ |
| medium | $\mathbf{231 \pm 1}$ | $\underline{232 \pm 2}$ |
| expert | $176 \pm 1$ | $\underline{208 \pm 2}$ |

Table 3: **MBRL experiment.** The PG loss with optimal reward comes second to the NLL baseline in terms of NLL, and ranks **first** in terms of MSE.

## 7 CONCLUSION

In this paper, we investigated how to learn reward functions that, when used within a policy gradient algorithm, produce models that are optimal in the sense of maximum likelihood with respect to observed data. To address this question, we introduced a bilevel optimization framework and derived closed-form solutions under specific assumptions on the reward model and the data-generating distribution. Finally, we validated our approach against practical applications, showing that our framework facilitates a more effective use of the advantages of PG methods through an optimal choice of the reward function.

**Limitations.** The reward parametrization considered in our work is relatively restrictive, which may affect its flexibility across different tasks. In addition, although we validated the framework on both synthetic and practical settings, further large-scale experiments are required to better understand its generalizability to more complex applications.

**Future directions.** While our experiments have so far focused on tabular data, we aim to broaden the scope to settings where MLE is known to face challenges such as compounding errors, exposure bias, and limited exploration. These include LLM fine-tuning, structured prediction tasks like machine translation, and time series forecasting. We plan to actively pursue this direction in future work.

## REPRODUCIBILITY STATEMENT

In order to ensure reproducibility we will release the code at `<URL hidden for review>`, once the paper has been accepted. Implementation details and relevant hyperparameters are provided in each experiment section of the main text.

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

# Appendix

## LLM Usage Disclosure

We used Large Language Models to help in the writing this paper, as well as in parts of the coding process. All outputs were reviewed and edited by the authors, who take full responsibility for the final content.

## Table of Contents

# A  THEORETICAL ANALYSIS

## A.1  PROOFS OF PROPOSITION A.2 AND COROLLARY 4.4

### A.1.1  A USEFUL LEMMA

This lemma will be useful to show to concavity of the objective function in the following proposition.

**Lemma A.1.** *Let* $(A, B, C) \in S_n^+ \times S_n^+ \times \mathbb{R}^{n \times n}$, *then one has that:*

$$\mathrm{Tr}(ACBC^\top) \geq 0.$$

*Proof.* Since $A$ is symmetric positive semidefinite, there exists a symmetric matrix $A^{1/2}$ such that

$$A = (A^{1/2})^2.$$

Then,

$$\mathrm{Tr}(ACBC^\top) = \mathrm{Tr}(A^{1/2}A^{1/2}CBC^\top).$$

Then, it follows that :

$$\mathrm{Tr}(A^{1/2}A^{1/2}CBC^\top) = \mathrm{Tr}(A^{1/2}CBC^\top A^{1/2}).$$

Let

$$M = A^{1/2}CBC^\top A^{1/2}.$$

Then,

$$\mathrm{Tr}(ACBC^\top) = \mathrm{Tr}(M).$$

So now it suffices to show that $M$ is definite semipositive.

The matrix $M$ is symmetric. For any vector $x \in \mathbb{R}^n$,

$$x^\top M x = x^\top A^{1/2}CBC^\top A^{1/2}x = (C^\top A^{1/2}x)^\top B(C^\top A^{1/2}x).$$

Since $B$ is positive semidefinite,

$$(C^\top A^{1/2}x)^\top B(C^\top A^{1/2}x) \geq 0.$$

Hence, $M$ is positive semidefinite. $\qquad\square$

### A.1.2  AN INTERMEDIATE PROPOSITION: SOLUTION OF THE INNER-LEVEL PROBLEM

We start by proving the following proposition on the closed-form solution of the inner level problem in Bi-O:

**Proposition A.2.** *Under Assumptions 4.1 and 4.2, the inner-level optimization problem*

$$\theta_U^\star = \arg\max_{\theta \in \Theta} \mathbb{E}_{X,Y \sim q}\Big[\mathbb{E}_{\widehat{Y}|X \sim \hat{p}_\theta}\Big[-(\widehat{Y} - Y)^T U(\widehat{Y} - Y)\Big] + \lambda\mathcal{H}(\hat{p}_\theta)\Big].$$

*admits exactly one solution that writes as*

$$\boxed{\theta^\star(U) = \left(\Lambda, \frac{\lambda U^{-1}}{2}\right).}$$

*Proof of Proposition A.2.* We prove the proposition by deriving a closed-form expression for the objective $\theta \mapsto J(\theta)$ and it's gradient, then we show that $J$ is strictly concave in $\theta = (A, B)$ over $\mathbb{R}^{n \times n} \times S_n^{++}(\mathbb{R})$ which guarantee that there is exactly one solution.

The objective function is:

$$J(\theta) = \mathbb{E}_X \mathbb{E}_{Y|X} \left[\mathbb{E}_{\widehat{Y} \sim P_\theta}\left[-(\widehat{Y} - Y)^T U(\widehat{Y} - Y) + \lambda H(\widehat{Y} \mid X)\right]\right].$$

First, we compute the inner expectation over $\widehat{Y}$ for fixed $X$ and $Y$. Since $\widehat{Y} \mid X \sim \mathcal{N}(AX, B)$, the entropy of $\widehat{Y} \mid X$ is:

$$H(\widehat{Y} \mid X) = \frac{1}{2}\log(2\pi e \det(B)).$$

Now, define:
$$I_{\widehat{Y}} := \mathbb{E}_{\widehat{Y} \sim P_\theta} \left[ -(\widehat{Y} - Y)^T U (\widehat{Y} - Y) + \lambda H(\widehat{Y} \mid X) \right].$$

$$I_{\widehat{Y}} = \underbrace{\mathbb{E}_{\widehat{Y} \sim P_\theta} \left[ -(\widehat{Y} - Y)^T U (\widehat{Y} - Y) \right]}_{\mathcal{A}} + \frac{\lambda}{2} \log(2\pi e \det(B)).$$

For fixed $X$ and $Y$, let $Z = \widehat{Y} - Y$. We show that :
$$\mathcal{A} = - \left[ X^T A^T U A X + \operatorname{Tr}(UB) - 2Y^T U(AX) + Y^T U Y \right].$$

Expanding the quadratic form:
$$Z^T U Z = \widehat{Y}^T U \widehat{Y} - 2Y^T U \widehat{Y} + Y^T U Y.$$

Taking expectations:
$$\mathbb{E}_{\widehat{Y}|X} \left[ \widehat{Y}^T U \widehat{Y} - 2Y^T U \widehat{Y} + Y^T U Y \right] = \mathbb{E}[\widehat{Y}^T U \widehat{Y}] - 2Y^T U \mathbb{E}[\widehat{Y}] + Y^T U Y.$$

Since $\widehat{Y} \mid X \sim \mathcal{N}(AX, B)$, we have $\mathbb{E}[\widehat{Y} \mid X] = AX$. Using the formula for the expectation of a quadratic form, for a random vector $W$ with mean $\mu$ and covariance $K$:
$$\mathbb{E}[W^T U W] = \mu^T U \mu + \operatorname{Tr}(UK).$$

Here, $W = \widehat{Y}$, $\mu = AX$, $K = B$, so:
$$\mathbb{E}[\widehat{Y}^T U \widehat{Y} \mid X] = (AX)^T U(AX) + \operatorname{Tr}(UB).$$

Thus, we get the desired expression for $\mathcal{A}$. It follows that,
$$I_{\widehat{Y}} = -X^T A^T U A X - \operatorname{Tr}(UB) + 2Y^T U A X - Y^T U Y + \frac{\lambda}{2} \log(2\pi e \det(B)).$$

Now, for fixed $X$, we compute:
$$J_X(A, B) = \mathbb{E}_{Y|X} \left[ I_{\widehat{Y}} \right].$$

Since $Y \mid X \sim \mathcal{N}(\Lambda X, \Sigma)$, we have $\mathbb{E}[Y \mid X] = \Lambda X$. Using quadratic form expectation again:
$$\mathbb{E}_{Y|X}[Y^T U Y] = X^T \Lambda^T U \Lambda X + \operatorname{Tr}(U\Sigma).$$

Thus,
$$J_X(A, B) = \mathbb{E}_{Y|X} \left[ -X^T A^T U A X - \operatorname{Tr}(UB) + 2Y^T U A X - Y^T U Y \right] + \frac{\lambda}{2} \log(2\pi e \det(B))$$

$$= -X^T A^T U A X - \operatorname{Tr}(UB) + 2\mathbb{E}_{Y|X}[Y]^T U A X - \mathbb{E}_{Y|X}[Y^T U Y] + \frac{\lambda}{2} \log(2\pi e \det(B))$$

$$= -X^T A^T U A X - \operatorname{Tr}(UB) + 2(\Lambda X)^T U A X - \left( X^T \Lambda^T U \Lambda X + \operatorname{Tr}(U\Sigma) \right) + \frac{\lambda}{2} \log(2\pi e \det(B))$$

$$= -X^T \left( A^T U A - 2\Lambda^T U A + \Lambda^T U \Lambda \right) X - \operatorname{Tr}(U(B + \Sigma)) + \frac{\lambda}{2} \log(2\pi e \det(B)).$$

Since $U$ is symmetric:
$$A^T U A - 2\Lambda^T U A + \Lambda^T U \Lambda = (A - \Lambda)^T U (A - \Lambda),$$

One has that:

$$J(\theta) = \mathbb{E}_X \left[ -X^T (A - \Lambda)^T U (A - \Lambda) X \right] - \operatorname{Tr}(U(B + \Sigma)) + \frac{\lambda}{2} \log(2\pi e \det(B))$$

Let $M_A = (A - \Lambda)^T U (A - \Lambda)$. it follows that (since $\operatorname{Tr}(x) = x$ for any $x \in \mathbb{R}$ and here $X^T M_A X \in \mathbb{R}$):
$$\mathbb{E}_X \left[ X^T M_A X \right] = \mathbb{E}_X \left[ \operatorname{Tr}(X^T M_A X) \right] = \mathbb{E}_X \left[ \operatorname{Tr}(M_A X X^T) \right] = \operatorname{Tr}(M_A \mathbb{E}_X[X X^T]).$$

Let $\Sigma_X = \mathbb{E}_X[XX^T]$. Thus,

$$\mathbb{E}_X\left[X^T M_A X\right] = \text{Tr}\left((A-\Lambda)^T U(A-\Lambda)\Sigma_X\right).$$

$$J(A,B) = -\,\text{Tr}\left((A-\Lambda)^T U(A-\Lambda)\Sigma_X\right) - \text{Tr}(U(B+\Sigma)) + \frac{\lambda}{2}\log(2\pi e \det(B)) \qquad (2)$$

Let $t \in \mathbb{R}$ and $u_1 = (A_1, B_1)$ and $u_2 = (A_2, B_2)$ such that $u_1 + tu_2 \in \mathbb{R}^{n\times n} \times S_n^{++}$. It suffices to show that $g : t \mapsto J(u_1 + tu_2)$ is a concave function on $N = \{t \in \mathbb{R}, \quad u_1 + tu_2 \in \mathbb{R}^{n\times n} \times S_n^{++}\}$.

Let $t \in N$, one has

$$u_1 + tu_2 = (A_1 + tA_2, B_1 + tB_2),$$

$$g(t) = -\underbrace{\text{Tr}(U\left(B_1 + tB_2 + U\Sigma\right) - \text{Tr}\left((A_1 + tA_2 - \Lambda)^\top U\left(A_1 + tA_2 - \Lambda\right)\Sigma_X\right)}_{:=H_1(t)}$$

$$+ \underbrace{\frac{\lambda}{2}\log\left(2\pi e \det(B_1 + tB_2)\right)}_{:=H_2(t)}.$$

First, $g \in C^2\left(N, \mathbb{R}\right)$.

Regarding $H_1$, a straightforward calculation shows that

$$t \mapsto H_1(t) = \alpha t^2 + \beta t + \gamma,$$

where:

$$\alpha = -\,\text{Tr}(U A_2 \Sigma_X A_2^\top),$$
$$\beta = -2\,\text{Tr}(U(A_1 - \Lambda)\Sigma_X A_2^\top) - \text{Tr}(UB_2),$$
$$\gamma = -\,\text{Tr}(U(A_1 - \Lambda)\Sigma_X(A_1 - \Lambda)^\top) - \text{Tr}(UB_1 + U^2\Sigma).$$

The second derivative is:
$$t \mapsto H_1''(t) = -2\,\text{Tr}(U A_2 \Sigma_X A_2^\top).$$

Since $U, \Sigma_X \in S_n^+(\mathbb{R})$, and $A_2 \in \mathbb{R}^{n\times n}$ we apply the lemma A.1 which gives us immediately that $\text{Tr}(U A_2 \Sigma_X A_2^\top) \geq 0$. Thus: $H_1$ is concave.

For $H_2$ one can show, using Jacobi's formulas that

$$\forall t \in N \quad H_2'(t) = \frac{d}{dt}\log(\det(B_1 + (\cdot)B_2)) = \text{Tr}\left((B_1 + tB_2)^{-1}B_2\right)$$

Since $B_1, B_2 \in S_n^{++}$ on can find two basis $\mathcal{B}_1$ and $\mathcal{B}_2$ such that they are diagonal in these bases,

$$\exists \boldsymbol{\lambda} = (\lambda_1, \ldots, \lambda_n) \in \mathbb{R}^n \setminus \{\mathbf{0}_n\}, \quad \exists \boldsymbol{\mu} = (\mu_1, \ldots, \mu_n) \in \mathbb{R}^n \setminus \{\mathbf{0}_n\}$$
$$\text{such that} \quad (B_1)_{\mathcal{B}_1} = \text{Diag}(\boldsymbol{\lambda}) \quad \text{and} \quad (B_2)_{\mathcal{B}_2} = \text{Diag}(\boldsymbol{\mu})$$

So

$$\forall t \in N \quad H_2'(t) = \sum_{1 \leq i \leq n} \frac{\mu_i}{\lambda_i + t\mu_i},$$

so

$$\forall t \in N \quad H_2''(t) = -\sum_{1 \leq i \leq n} \frac{\mu_i^2}{(\lambda_i + t\mu_i)^2} < 0.$$

So $g$ is a sum of a concave and a strictly concave function so it's a strictly concave function and thus $J$ is strictly concave, thus the problem A.2 admits exactly one solution on $\mathbb{R}^{n\times n} \times S_n^{++}(\mathbb{R})$; which is solution of

$$\nabla J(A,B) = 0. \qquad (3)$$

Let's find in closed form the solutions of the previous equation.

It follows that:
$$\nabla_A J(A, B) = -2U(A - \Lambda)\Sigma_X$$

Then,

$$\nabla_B J(A, B) = -U^T + \frac{\lambda}{2}\frac{\partial \ln(\det)(B)}{\partial B} = (B^{-1})^T = -U + \frac{\lambda}{2}B^{-1}$$

Finally:

$$\boxed{\theta^\star(U) = \left(\Lambda, \frac{\lambda U^{-1}}{2}\right).}$$

$\square$

### A.1.3 PROOF OF PROPOSITION 4.3

We restate the proposition before proceeding with the proof:

**Proposition A.3.** *Given the above assumptions and the solution in Proposition A.2, the outer-level problem*

$$U^\star = \underset{U \in S_n^{++}(\mathbb{R})}{\arg\min}\ \mathbb{E}_{X,Y\sim q}\big[\log \hat{p}_{\theta_U^\star}(Y|X)\big],$$

*has at least one solution:*

$$\boxed{U^\star = \frac{\lambda \Sigma^{-1}}{2}.}$$

*Proof.* Let $U \in S_n^{++}(\mathbb{R})$ and $(\lambda, n) \in \mathbb{R}_+^\star \times \mathbb{N}^\star$ by the previous proposition one has $\theta^\star(U) = \left(\Lambda, \frac{\lambda U^{-1}}{2}\right).$

Let's check that

$$\varphi : U \mapsto \mathbb{E}_X D_{\mathrm{KL}}\big(q(\cdot \mid X)\,\|\,p_{\theta_U^\star}(\cdot \mid X)\big)$$

is a convex function of $U$.

To show the convexity of $\varphi$, we show the convexity of $g$, which is defined as follow. Let $U, V \in S_n^{++}(\mathbb{R})$ and $t \in I_{U,V} := \{u \in \mathbb{R}\quad U + uV \in S_n^{++}(\mathbb{R})\}$. Define

$$\forall t \in I_{U,V}\quad g(t) = \varphi(tU + V).$$

One has that $g \in C^2\left(I_{U,V}, \mathbb{R}\right)$.

Since both $q(\cdot \mid X)$ and $p_{\theta_U^\star}(\cdot \mid X)$ are Gaussian with the same mean $\Lambda X$, the Kullback–Leibler divergence has a closed-form expression:

$$D_{\mathrm{KL}}\big(q\,\|\,p_{\theta_U^\star}\big) = \frac{1}{2}\left[\mathrm{Tr}\left(\Sigma_p^{-1}\Sigma\right) - n + \ln\left(\frac{\det(\Sigma_p)}{\det(\Sigma)}\right)\right],$$

where $\Sigma_{p_{\theta_U^\star}} = \frac{\lambda}{2}U^{-1}$. It follows that,

$$\Sigma_{p_{\theta_U^\star}}^{-1} = \frac{2}{\lambda}U, \quad \text{and} \quad \det(\Sigma_{p_{\theta_U^\star}}^{-1}) = \left(\frac{\lambda}{2}\right)^n \det(U)^{-1}.$$

Substituting these in, we find:

$$D_{\mathrm{KL}}\big(q\,\|\,p_{\theta_U^\star}\big) = \frac{1}{2}\left[\mathrm{Tr}\left(\frac{2}{\lambda}U\Sigma\right) - n + \ln\left(\frac{(\lambda/2)^n}{\det(U)\det(\Sigma)}\right)\right]$$

$$= \frac{1}{\lambda}\mathrm{Tr}(U\Sigma) - \frac{n}{2} + \frac{n}{2}\ln\left(\frac{\lambda}{2}\right) - \frac{1}{2}\ln(\det(\Sigma)) - \frac{1}{2}\ln(\det(U)).$$

This expression is independent of $X$, so its expectation is itself:

$$\varphi(U) = \frac{1}{\lambda}\mathrm{Tr}(U\Sigma) - \frac{1}{2}\ln(\det(U)) + C,$$

where $C$ is a constant independent of $U$.

Now, we express $g(t)$ explicitly and set $\forall t \in I_{U,V} \quad A(t) = U + tV$:

$$\forall t \in I_{U,V} \quad g(t) = \varphi(A(t)) = \frac{1}{\lambda} \operatorname{Tr}(A(t)\Sigma) - \frac{1}{2} \ln(\det(A(t))) + C.$$

Clearly, $g \in C^2(I_{U,V}, \mathbb{R})$, let's show that its second derivative is positive.

The first derivative is:

$$\forall t \in I_{U,V}, \quad g'(t) = \frac{1}{\lambda} \operatorname{Tr}(U\Sigma) - \frac{1}{2} \frac{d}{dt} \ln(\det(A(t))).$$

Using the identity that follows from Jacobi's formula:

$$\forall t \in I_{U,V}, \quad \frac{d}{dt} \ln(\det(A(t))) = \operatorname{Tr}\left(A(t)^{-1}A'(t)\right),$$

we get:

$$\forall t \in I_{U,V}, \quad g'(t) = \frac{1}{\lambda} \operatorname{Tr}(U\Sigma) - \frac{1}{2} \operatorname{Tr}(A(t)^{-1}U).$$

Differentiating again, one has that:

$$\forall t \in I_{U,V}, \quad g''(t) = -\frac{1}{2} \frac{d}{dt} \operatorname{Tr}(A(t)^{-1}U).$$

Therefore,

$$\forall t \in I_{U,V}, \quad g''(t) = \frac{1}{2} \operatorname{Tr}(A(t)^{-1}U A(t)^{-1}U).$$

Let $t \in I_{U,V}$ and $B = A(t)^{-1/2}U A(t)^{-1/2}$, where $A(t)^{1/2}$ is the symmetric positive definite square root of $A(t)$. Since $U$ is positive definite, $B$ is also positive definite. We have:

$$\begin{aligned}
\operatorname{Tr}(A(t)^{-1}U A(t)^{-1}U) &= \operatorname{Tr}(A(t)^{-1/2}A(t)^{-1/2}U A(t)^{-1/2}A(t)^{-1/2}U) \\
&= \operatorname{Tr}(A(t)^{-1/2}U A(t)^{-1/2}A(t)^{-1/2}U A(t)^{-1/2}) \\
&= \operatorname{Tr}(BB) = \operatorname{Tr}(B^2).
\end{aligned}$$

Thus,

$$g''(t) = \frac{1}{2} \operatorname{Tr}(B^2).$$

Since $B$ is symmetric and positive definite, its eigenvalues $\lambda_1, \ldots, \lambda_n$ are positive. Therefore,

$$\operatorname{Tr}(B^2) = \sum_{i=1}^{n} \lambda_i^2 > 0,$$

which implies $g''(t) > 0$ for all $t \in I_{U,V}$.

Since the second derivative of $g$ is strictly positive on its domain, $g$ is strictly convex. Thus $\varphi$ is strictly convex on $S_n^{++}(\mathbb{R})$.

The existence and the uniqueness is shown.

By the moment matching principle for Kullback–Leibler divergence one find that

$$\boxed{U^\star = \frac{\lambda \Sigma^{-1}}{2}.}$$

$\square$

### A.1.4  PROOF OF COROLLARY 4.4

**Corollary A.4** (Isotropic case). *If we assume that,* $\mathrm{B} = \sigma^2 \mathrm{I}_n{}^2$, *the set of solutions to the outer-level problem is characterized by:*

$$\boxed{\mathrm{U}^\star \in F_{\lambda,\Sigma} := \left\{ \mathrm{U} \in S_n^{++}(\mathbb{R}) \,\middle|\, \mathrm{Tr}(\mathrm{U}) = \frac{\lambda n^2}{2\,\mathrm{Tr}(\Sigma)} \right\}.}$$

*Proof.* The proof follows the same calculations and arguments as those used in the proof of Proposition A.2 and Proposition 4.3. Specifically, we show that the objective function $(A, \sigma^2) \mapsto J(A, \sigma^2)$ is concave in $(A, \sigma^2)$ and solve the first-order optimality conditions. This leads to the solution

$$\theta^\star(U) = \left( \Lambda, \frac{\lambda n}{2\,\mathrm{Tr}(U)} \right).$$

Substituting this solution into the $\mathrm{D}_{KL}$ expression and following the same arguments leads to $F_{\lambda,\Sigma}$. $\qquad\square$

### A.1.5  PROOF OF COROLLARY 4.5

*Proof.* Substituting $\mathrm{U}^\star = \frac{\lambda}{2}\Sigma^{-1}$ gives:

$$J(\theta) = \mathbb{E}_X \mathbb{E}_{Y|X \sim q} \left[ \left[ \mathbb{E}_{\widehat{Y} \sim \hat{p}_\theta(\cdot|X)} \left[ -\frac{\lambda}{2}(\widehat{Y} - Y)^\top \Sigma^{-1}(\widehat{Y} - Y) \right] \right] \right] + \lambda \mathbb{H}(\hat{p}_\theta).$$

Since $q(Y|X)$ is Gaussian with mean $AX$ and covariance $\Sigma$, write $Y = AX + \varepsilon$ with $\varepsilon \sim \mathcal{N}(0, \Sigma)$. Remark that:
$$\widehat{Y} - Y = (\widehat{Y} - AX) - \varepsilon.$$

Thus,

$$(\widehat{Y} - Y)^\top \Sigma^{-1}(\widehat{Y} - Y) = (\widehat{Y} - AX)^\top \Sigma^{-1}(\widehat{Y} - AX) - 2(\widehat{Y} - AX)^\top \Sigma^{-1}\varepsilon + \varepsilon^\top \Sigma^{-1}\varepsilon.$$

Taking the conditional expectation $\mathbb{E}_{Y|X}$:

$$\begin{aligned}
\mathbb{E}_{Y|X} \left[ (\widehat{Y} - Y)^\top \Sigma^{-1}(\widehat{Y} - Y) \right] &= (\widehat{Y} - AX)^\top \Sigma^{-1}(\widehat{Y} - AX) - 0 + \mathbb{E}[\varepsilon^\top \Sigma^{-1}\varepsilon] \\
&= (\widehat{Y} - AX)^\top \Sigma^{-1}(\widehat{Y} - AX) + \mathrm{tr}(\Sigma^{-1}\Sigma) \\
&= (\widehat{Y} - AX)^\top \Sigma^{-1}(\widehat{Y} - AX) + n,
\end{aligned}$$

where $n \in \mathbb{N}^\star$ is the dimension of $Y$.

Therefore,

$$\mathbb{E}_{Y|X} \left[ -\frac{\lambda}{2}(\widehat{Y} - Y)^\top \Sigma^{-1}(\widehat{Y} - Y) \right] = -\frac{\lambda}{2} \left[ (\widehat{Y} - AX)^\top \Sigma^{-1}(\widehat{Y} - AX) + n \right].$$

Now, the log-likelihood of $\widehat{Y}$ under $q(\cdot|X)$ is:

$$\log q(\widehat{Y}|X) = -\frac{1}{2}(\widehat{Y} - AX)^\top \Sigma^{-1}(\widehat{Y} - AX) - \frac{1}{2} \log\left( (2\pi)^n |\Sigma| \right).$$

Thus,

$$(\widehat{Y} - AX)^\top \Sigma^{-1}(\widehat{Y} - AX) = -2\log q(\widehat{Y}|X) - \log\left( (2\pi)^n |\Sigma| \right).$$

one has:

$$\mathbb{E}_{Y|X} \left[ -\frac{\lambda}{2}(\widehat{Y} - Y)^\top \Sigma^{-1}(\widehat{Y} - Y) \right] = -\frac{\lambda}{2} \left[ -2\log q(\widehat{Y}|X) - \log\left( (2\pi)^n |\Sigma| \right) + n \right]$$

---

[2]we denote by $\mathrm{I}_n$ the identity matrix of size $n$.

$$= \lambda \log q(\widehat{Y}|X) + \underbrace{\frac{\lambda}{2} \left[ \log \left( (2\pi)^n |\Sigma| \right) - n \right]}_{:=c_n}.$$

The term $c_n$ is constant with respect to $\widehat{Y}$ and $\theta$. Therefore,

$$\mathbb{E}_{X \sim q} \left[ \mathbb{E}_{\widehat{Y} \sim \hat{p}_\theta(\cdot|X)} \left[ \mathbb{E}_{Y|X} \left[ -\frac{\lambda}{2} (\widehat{Y} - Y)^\top \Sigma^{-1} (\widehat{Y} - Y) \right] \right] \right] = \lambda \mathbb{E}_{X \sim q} \left[ \mathbb{E}_{\widehat{Y} \sim \hat{p}_\theta(\cdot|X)} \left[ \log q(\widehat{Y}|X) \right] \right] + c_n$$

The entropy term is:

$$\lambda \mathbb{H}(\hat{p}_\theta) = \lambda \mathbb{E}_{X \sim q} \left[ \mathbb{E}_{\widehat{Y} \sim \hat{p}_\theta(\cdot|X)} \left[ -\log \hat{p}_\theta(\widehat{Y}|X) \right] \right].$$

Thus, the objective function becomes:

$$J(\theta) = \lambda \mathbb{E}_{X \sim q} \left[ \mathbb{E}_{\widehat{Y} \sim \hat{p}_\theta(\cdot|X)} \left[ \log q(\widehat{Y}|X) - \log \hat{p}_\theta(\widehat{Y}|X) \right] \right] + c_n$$
$$= -\lambda \mathbb{E}_{X \sim q} \left[ d_{\mathrm{KL}} \left( \hat{p}_\theta(\cdot|X) \, \| \, q(\cdot|X) \right) \right] + c_n.$$

So, maximizing $J(\theta)$ is equivalent to minimizing the reverse KL divergence, which completes the proof. $\qquad\square$

### A.2 ON THE DEFINITION OF $\mathcal{H}$ IN *Assumption* 4.2

Let

$$\mathcal{H} := \left( L^2 \left( \mathbb{R}^n \times \mathbb{R}^n, \, \mathbb{R}, \, e^{-\|X - X'\|^2} \, d\lambda(X, X') \right) ; \, \langle \cdot, \cdot \rangle_{\mathcal{H}} \right),$$

where

$$\forall f, g \in \mathcal{H} \quad \langle f, g \rangle_{\mathcal{H}} = \int_{\mathbb{R}^n \times \mathbb{R}^n} f(X, X') \, g(X, X') \, e^{-\|X - X'\|^2} \, d\lambda(X) \, d\lambda(X')$$

and $d\lambda$ denotes the Lebesgue measure.

**Lemma A.5.** *Let $U \in \mathbb{R}^{n \times n}$, then $r_U$ as defined in 4.2 is an element of*

$$\mathcal{H} := L^2 \left( \mathbb{R}^n \times \mathbb{R}^n, \, \mathbb{R}, \, e^{-\|X - X'\|^2} \, d\lambda(X, X') \right).$$

*Proof.* We denote by $\langle \cdot, \cdot \rangle_{\mathbb{R}^n}$ the usual Euclidean scalar product on $\mathbb{R}^n$. In particular, for any $X, X' \in \mathbb{R}^n$ and any matrix $U \in \mathbb{R}^{n \times n}$, we have by the Cauchy-Schwarz inequality

$$|(X - X')^\top U (X - X')| = |\langle X - X', U(X - X') \rangle_{\mathbb{R}^n}| \leq \|X - X'\|^2 \cdot \|U(X - X')\|^2 \quad (4)$$

and notice that for any $U \in \mathbb{R}^{n \times n}$ and $X \in \mathbb{R}^n$,

$$\|UX\|^2 = \sum_{1 \leq k \leq n} \left( \sum_{j=1}^n U_{k,j} X_j \right)^2 \leq n^2 \underbrace{\left( \max_{(k,j) \in [1,n]^2} |U_{k,j}|^2 \right)}_{:=C(n,U) > 0} \|X\|^2. \quad (5)$$

It leads to:

$$\int_{\mathbb{R}^n \times \mathbb{R}^n} \left( -(X - X')^\top U (X - X') \right)^2 e^{-\|X - X'\|^2} \, d\lambda(X, X')$$

$$\underbrace{\leq}_{(4)} \int_{\mathbb{R}^n \times \mathbb{R}^n} \|X - X'\|^2 \cdot \|U(X - X')\|^2 e^{-\|X - X'\|^2} \, d\lambda(X, X')$$

$$\underbrace{\leq}_{(5)} C(n, U) \int_{\mathbb{R}^n \times \mathbb{R}^n} \|X - X'\|^4 e^{-\|X-X'\|^2} \, d\lambda(X, X')$$

$$< \infty.$$

Since;

$$\int_{\mathbb{R}^n \times \mathbb{R}^n} \|X - X'\|^4 e^{-\|X-X'\|^2} d\lambda(X, X') = \int_{\mathbb{R}^n} \|Z\|^4 e^{-\|Z\|^2} dZ$$

$$= \text{Vol}(S^{n-1}) \int_0^\infty r^{n-1} \cdot r^4 e^{-r^2} dr < \infty.$$

$$r_U \in \mathcal{H}.$$

$\square$

The following two lemmas justifies the reparametrization search space by $S_n^{++}(\mathbb{R})$.

**Lemma A.6.** *The set $\{r_U \in \mathcal{H} \quad : U \in S_n^{++}(\mathbb{R})\}$ is in bijection with $S_n^{++}(\mathbb{R})$.*

*Proof.* Denote $I := \{r_U \in \mathcal{H} \quad : U \in S_n^{++}(\mathbb{R})\}$ and

$$\varphi : S_n^{++} \to I$$

defined by

$$\forall U \in S_n^{++} \quad \varphi(U) = r_U.$$

The surjectivity of $\varphi$ is straightforward since the image of $\varphi$ is $I$. Let $U_1, U_2 \in S_n^{++}$ and assume $\phi(U_1) = \phi(U_2)$, which is

$$\forall X, Y \in \mathbb{R}^n, \quad (X - Y)^T (U_1 - U_2)(X - Y) = 0. \tag{6}$$

But one can find $P \in GL_n(\mathbb{R})$ such that

$$U_1 - U_2 = PDP^T, \quad D = \text{diag}(\lambda_1, \ldots, \lambda_n).$$

Then (6) reads:

$$\forall W = (w_1, ..., w_n) \in \mathbb{R}^n \quad \sum_{i=1}^n \lambda_i \underbrace{w_i^2}_{\geq 0} = 0.$$

Thus $\forall i \in [1, n] \quad \lambda_i = 0$, so $U_1 = U_2$. $\square$

**Lemma A.7.** *Let $X$ and $Y$ be two non-empty sets and let $\phi : X \to Y$ be a bijection. Let $f : X \to \mathbb{R}$ and $g : Y \to \mathbb{R}$, such that*

$$\forall x \in X \quad g(\phi(x)) = f(x) \tag{7}$$

*Then the optimization problems:*

$$\text{(P1)} \quad \max_{x \in X} f(x) \quad \text{and} \quad \text{(P2)} \quad \max_{y \in Y} g(y),$$

*are equivalent in the sense that:*

- *If $x^*$ is a solution of (P1), then $y^* = \phi(x^*)$ is a solution of (P2).*

- *Conversely, if $y^*$ is a solution of (P2), then $x^* = \phi^{-1}(y^*)$ is a solution of (P1).*

*Proof.* First; let's show that (7) implies that

$$\forall y \in Y \quad f(\phi^{-1}(y)) = g(y). \tag{8}$$

Let $y \in Y$, since $\phi$ is a bijection one can find $x \in X$ such that $y = \phi(x)$ and $x = \phi^{-1}(y)$ so using (7):

$$f \circ \phi^{-1}(y) = f(x) \underbrace{=}_{(7)} g \circ \phi(x) \underbrace{=}_{\text{bijectivity of } \phi} g \circ \phi \circ \phi^{-1}(y) = g(y). \tag{9}$$

Let's now proove that the sup of the two problems are equals.

Since $\phi$ is a bijection, every element $y \in Y$ can be uniquely written as $y = \phi(x)$ for some $x \in X$. By assumption, we then have $g(y) = g(\phi(x)) = f(x) \leq \sup_{x \in X} f(x) := M_f$.

So
$$\sup_{y \in Y} g(y) := M_g \leq M_f \tag{10}$$

Let $x \in X$, by the bijection, one can find $y \in Y$ such that $x = \phi^{-1}(y)$ so $f(x) = f(\phi^{-1}(y)) = g(y) \leq M_g$.

It follows that
$$M_f \leq M_g \tag{11}$$

Combining (10) and (11):
$$\sup_{y \in Y} g(y) = \sup_{x \in X} f(x).$$

Furthermore, if $x^*$ is a point where $f$ attains its maximum, then for $y^* = \phi(x^*)$, we have:
$$g(y^*) = f(x^*) = \max_{x \in X} f(x) = \max_{y \in Y} g(y),$$

so $y^*$ is a solution of (P2). Conversely, if $y^*$ is a point where $g$ attains its maximum, then let $x^* = \phi^{-1}(y^*)$. We have:
$$f(x^*) = g(\phi(x^*)) = g(y^*) = \max_{y \in Y} g(y) = \max_{x \in X} f(x),$$

so $x^*$ is a solution of (P1). $\qquad\square$

**Proposition A.8.** *For each $U \in S_n^{++}(\mathbb{R})$, define*
$$f(U) = \mathbb{E}_X \mathbb{E}_{Y|X \sim q}[\log \hat{p}_{\theta_U}(Y \mid X)].$$

*For each function $r \in I$, where*
$$I = \left\{ r_U : U \in S_n^{++}(\mathbb{R}) \right\}, \qquad with \qquad r_U(\widehat{Y}, Y) = -(\widehat{Y} - Y)^\top U(\widehat{Y} - Y),$$

*define*
$$g(r) = \mathbb{E}_X \mathbb{E}_{Y|X \sim q} \log \hat{p}_{\theta_r}(Y \mid X).$$

*Then the optimization problems*
$$\max_{U \in S_n^{++}(\mathbb{R})} f(U) \qquad and \qquad \max_{r \in I} g(r)$$

*are equivalent.*

*Proof.* It's a straightforward consequence of the lemma A.7 with $X := S_n^{++}(\mathbb{R})$ and $Y := I$ and the map $\phi : X \to Y$ defined by $\phi(U) = r_U$, for which we now that, by Lemma A.6, is a bijection.

$\qquad\square$

# B ADDITIONAL EXPERIMENTS

## B.1 DISTRIBUTION COMPARISON

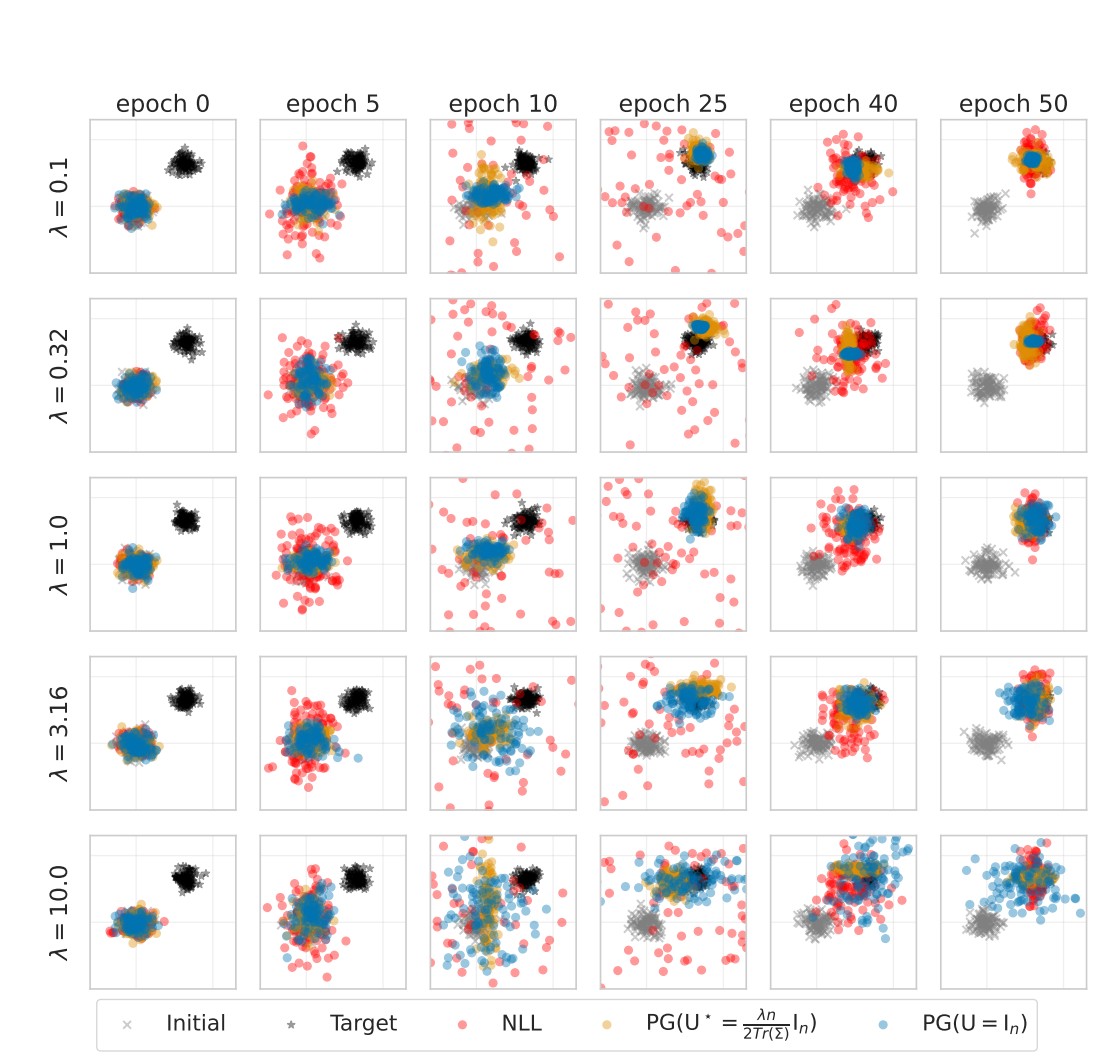

Figure 4: Distribution comparison, different value of $\lambda$

