# OpenReview forum: "From Data to Rewards: a Bi-level Optimization Perspective on Maximum Likelihood Estimation"
_ICLR.cc/2026/Conference — Submitted to ICLR 2026_

### Official Review · Reviewer_ZjsV · 2025-10-25

**Soundness:** 3
**Presentation:** 2
**Contribution:** 2
**Rating:** 4
**Confidence:** 4

**Summary:**

This paper proposes a bilevel optimization (Bi-O) framework that reinterprets Maximum Likelihood Estimation (MLE) for generative models as a reward-learning problem. The key idea is to treat the reward function as an outer-level optimization variable and the policy gradient (PG) objective as an inner-level problem.
The authors show that under Gaussian assumptions, the Bi-O problem admits a closed-form solution. Then, they generalize to applications such as tabular classification and model-based reinforcement learning.

**Strengths:**

- The paper is generally well-written. The mathematical derivations are well organized.
- Conceptual novelty: Provides a unifying view linking MLE and RL-based training through a bilevel optimization formulation.

**Weaknesses:**

- Experiments are restricted to small-scale synthetic and tabular datasets. It’s unclear how the method scales to large generative models (e.g., language models). Although the paper motivates connections to LLM training (e.g., RLHF), the experiments do not test any large-scale or sequential generative task.
- The Gaussian assumption (Assumption 4.1) limits generality; it’s not clear how the conclusions extend beyond this setting.
- The reward model (Assumption 4.2) is in a limited form. In many real-world applications, the reward function does not take such quadratic form, for example, in recent LLM RL training such as in (Shao et al., 2024; DeepSeek-AI et al., 2025).
- The expriments in 6.1 (TABULAR CLASSIFICATION) and 6.2 (MODEL-BASED REINFORCEMENT LEARNING) are not detailed. The presentation should be improved so that the readers can more easily grasp the important part of developments and results related to real-world applications. Instead, the space for introducing the Gaussian example can be shortened.

**Questions:**

see above.

---

> ### Author Response · Authors · 2025-11-19
> **Rebuttal by Authors**
>
> We thank Reviewer ZjsV for their review and the time dedicated to evaluating our work. We appreciate their suggestions and will take them into account as we revise and continue to improve the paper.

---

### Official Review · Reviewer_pYjD · 2025-10-27

**Soundness:** 2
**Presentation:** 2
**Contribution:** 2
**Rating:** 4
**Confidence:** 3

**Summary:**

The paper proposes a principled way to turn supervised data into an RL-style reward so that a policy-gradient (PG) inner loop can be used while still targeting a maximum-likelihood (MLE) outer objective. Concretely, MLE is recast as a bilevel optimization where the outer problem optimizes a reward function $r$, and the inner problem optimizes model parameters $\theta$ via an entropy-regularized PG objective.

**Strengths:**

- The Gaussian/quadratic analysis yields a closed-form U and a clear reverse-KL equivalence, offering intuition for why PG with the right reward can mimic (or complement) MLE.
- The problem setup and contribution bullets are clear; figures for synthetic experiments.

**Weaknesses:**

- The main theorem relies on Gaussian conditionals and a quadratic (Mahalanobis) reward; it is unclear how sensitive the conclusions are when these assumptions fail.
- The paper said it aims at providing a general framework (i.e. beyond control tasks) than Zeng et. al. I am winding how such bi-level optimization perform in LLM setting. For example, a simple comparison with LLM's reasoning task using grpo/ppo with verifiable reward and some open-end domains with non-verifiable reward. So that we can see the practical usage of this bi-level optimization algorithm in different cases.

**Questions:**

same as weakness.

---

> ### Author Response · Authors · 2025-11-19
> **Rebuttal by Authors**
>
> We thank Reviewer pYjD for their review and the time dedicated to evaluating our work. We appreciate their suggestions and will take them into account as we revise and continue to improve the paper.

---

### Official Review · Reviewer_VZ8W · 2025-11-04

**Soundness:** 3
**Presentation:** 3
**Contribution:** 2
**Rating:** 2
**Confidence:** 4

**Summary:**

The authors propose to reframe maximum likelihood estimation as a bi-level optimization problem. They perform theoretical analysis of doing so in the Gaussian setting, before performing experiments on toy RL problems.

**Strengths:**

N/A

**Weaknesses:**

(-) I thought about this for a while and I am fairly convinced this paper is re-inventing Ziebart's classical MaxEnt IRL paper. Looking at the core optimization problem (Bi-O), we see it is set up as finding a reward function whose soft optimal policy has a high likelihood of generating the observed data. This is precisely what MaxEnt IRL is.

To compute the likelihood gradient of the reward function (which requires knowing the soft optimal policy), Ziebart et al. propose computing the policy via soft value iteration. I've read this paper a few times but I still can't tell what the authors are doing instead. I could there being see a difference between the classical formulation and this work here. However, it should be made explicit that this is the novel contribution, rather than the formulation itself.

(-) I would add in a reference to Ziebart's MaxEnt IRL paper above the definition of PG. Similarly, I'd add in a reference to Swamy's "All Roads Lead to Likelihood" paper re: MaxEnt RL minimizing reverse KL (Corr. 4.5).

(-) The experiments are limited and performed on toy domains. There is almost no connection between the theory and the practice

**Questions:**

(1) Can you spell out in greater detail what you're doing for your MBRL experiments?

---

> ### Author Response · Authors · 2025-11-19
> **Rebuttal by Authors**
>
> We thank Reviewer VZ8W for their review and the time dedicated to evaluating our work. We appreciate their suggestions and will take them into account as we revise and continue to improve the paper.
>
> Regarding the MBRL experiment, the task is to maximize the log-likelihood of the observed next states $\mathrm{Y} := \mathrm{S}_{t+1}$ conditioned on the current states and actions $\mathrm{X} := (\mathrm{S}_t,\mathrm{A}_t)$. Pluging-in this problem in the Bilevel Optimization Problem (Bi-O) and using the algorithm 1 and 2 to find the optimal matrix $\mathrm{U}^\star$ yields the results shown in Table 3.

---

### Meta-Review · Area_Chair_mJVL · 2025-12-13

**Summary:**

Reviewer scores are uniformly low. The main concerns are limited novelty relative to prior work, restrictive theoretical assumptions, and weak experimental validation. The rebuttal does not provide informed or substantive responses to these issues.

**Reviewer Concerns:**

Please see summary.

**Reviewer Scores:**

The rebuttal did not provide informed or substantive responses to the reviewers’ concerns, and therefore no score changes would be expected.

---

### Decision · Program_Chairs · 2026-01-26

Reject